# Population Subdivision and Migration Assessment of Mangalica Pig Breeds Based on Pedigree Analysis

**DOI:** 10.3390/ani14040653

**Published:** 2024-02-19

**Authors:** Anh Thi Nguyen, György Kövér, Péter Tóth, Ino Curik, Árpád Bokor, István Nagy

**Affiliations:** 1Institute of Animal Sciences, Hungarian University of Agriculture and Life Sciences (MATE), Guba Sándor u. 40, 7400 Kaposvár, Hungary; nguyen.thi.anh@phd.uni-mate.hu (A.T.N.); kover.gyorgy@uni-mate.hu (G.K.); bokor.arpad@uni-mate.hu (Á.B.); 2Department of Veterinary Medicine, Faculty of Biology Applied Sciences, Vinh Long University of Technology Education, 73 Nguyen Hue, Ward 2, Vinh Long 85000, Vietnam; 3Hungarian National Association of Mangalica Breeders, Piac u. 77, 4025 Debrecen, Hungary; toth.peter@moe.org.hu; 4Department of Animal Science, Faculty of Agriculture, University of Zagreb, Svetošimunska cesta 25, 10000 Zagreb, Croatia; icurik@agr.hr

**Keywords:** Mangalica pigs, population subdivision, Wright’s *F_ST_* coefficient, pedigree analysis, migration

## Abstract

**Simple Summary:**

The presence of substructures within isolated subpopulations can increase the risk of inbreeding. This ultimately leads to a reduction in overall genetic diversity and increased susceptibility to disease or other environmental stressors. Small subpopulations are more susceptible to genetic drift, where random events can lead to significant changes in genetic composition. This can be problematic for conservation measures aimed at preserving certain genetic traits. The Hungarian Mangalica with three different colour variants (Blonde, Red, Swallow-Bellied), representing three different breeds, have maintained their genetic and phenotypic appearance unchanged since 1976. As all breeds have been kept in multiple herds over a long period of time, the assessment of population subdivision based on these herds could help in investigating the dynamic change in population structure associated with conservation. In our study, the population substructure of each breed was evaluated using the concept of Wright-F statistics, and the results were presented using graphical methods (heat maps). The results showed that none of the breeds analysed had a composition of substructure. This favourable phenomenon is the result of an adequate migration among the herds, which proves the appropriateness of the applied breeding programme.

**Abstract:**

In conserving the genetic diversity of domestic animal breeds, strategies that emphasise between-breed diversity may not be optimal, as they neglect within-breed variation. The aim of the present study was to assess the extent of population subdivision in three Mangalica pig breeds and the contribution of migration to their substructure. Wright’s *F_ST_* coefficient was calculated based on genealogical data with breeding animals born between 1981 and 2023, with three colour variants (Blonde, Swallow-Belly and Red). These Wright’s *F_ST_* coefficients were analysed using multidimensional scaling to reveal the population substructure. The average *F_ST_* coefficient was 0.04 for the Blonde breed and 0.047 for the Swallow-Belly and Red Mangalica breeds, while these parameters were lower in the active herds at 0.03 and 0.04, respectively. The migration of individuals between herds was 61.63% for the Blonde breed and 75.53% and 63.64% for the Swallow-Belly and Red Magalica breeds, respectively. No population substructure was observed in any of the Mangalica breeds, which can be explained by the extensive migration between herds.

## 1. Introduction

In the 1830s, the Serbian Sumadia pig breed was crossed with the local Hungarian stock, and a rustic, curly pig called Blonde Mangalica [1] was created through intensive selection. Later, the Swallow-Belly Mangalica breed was created by crossing Mangalica pigs and Szerémségi pigs. The most recent breed is the Red Mangalica pig, which was created by crossing Mangalica pigs with Szalontai pigs and by crossing Újszalontai pigs with Mangalica pigs at the beginning of the 19th century [2]. Mangalica pigs are characterised by excellent fat production, strong maternal tendencies and a good adaptability to extensive husbandry conditions, but their reproductive capacity is low [1]. The Mangalica pig was the most important Hungarian pig breed until the 1950s. After the Second World War, the Mangalica lost its former popularity due to changes in dietary habits [3]. Although a national programme for preserving the gene pool was launched in 1976, the Mangalica breed was almost extinct by the beginning of 1990 [4]. Fortunately, the National Association of Mangalica Breeders was established in 1994 to preserve the genetic and phenotypic appearance of the Mangalica pig in an unaltered form [2]. Thanks to their efficient activity, the number of registered sows and boars (of the three breeds combined) in 2019 was 6723 and 354, respectively [5]. Currently, there are three different colour variants in the Mangalica (Blonde, Red, Swallow-Belly), and based on the molecular genetic analysis of Zsolnai et al. [6], it was found that these colour variants represent different breeds. In terms of gene conservation, Mangalica pig breeds are among the most recognised breeds in Hungary, which is why the conservation of these breeds is of great importance. However, in terms of breed loss, it is necessary to study the population substructure and migration in order to make an appropriate assessment of the management and conservation of genetic variability [7]. Recently, all Mangalica breeds were evaluated using pedigree analysis, which revealed the demographic parameters, the degree of inbreeding and the proportion of genetic diversity conserved [8]. However, as all breeds are kept in several herds, these herds can be interpreted as subpopulations.

The aim of the present study was to evaluate if the different Mangalica populations show any signs of subdivision and to determine the characteristics of migration in these populations.

## 2. Materials and Methods

Because the current study exclusively involved the analysis of genealogical data stored in datasets, approval from the Animal Care and Use Committee was not needed.

### 2.1. Genealogical Data

The data used for the research in this study was supported by the Hungarian National Association of Mangalica Breeders. The organisation documented information on registered Mangalica pigs listed in the Herdbook, consisting of pigs born between 1981 and 2023. The genealogy analysis was limited to Blonde, Swallow-Belly and Red Mangalica breeding animals (i.e., animals that have sired offspring), as in Table 1.

Due to the presence of numerous inactive herds (herds abandoned their breeding activity before 2023) during the investigation, the research was conducted both on the total herds and on the currently active herds, respectively.

### 2.2. Population Subdivision

The genealogical data were used to analyse the structure of the subpopulations using the concept of Wrigth’s F-statistics [9], calculated according to Caballero and Toro [10] for each specified subpopulation. As described in Gutiérrez and Goyache (2005) [11], we first calculated the average pairwise coancestry coefficient (fij) between individuals from two distinct subpopulations labelled i and j. The analysis includes all possible pairs of individuals within the entire metapopulation, taking into account the size of these subpopulations, so that the total of NixNj pairs are considered. According to Caballero and Toro [10], pedigree-based calculations assume that all coancestries are known through genealogical information back to the base population, in which all individuals are unrelated. Within a given subpopulation, labelled i, the following metrics can be calculated: the average coancestry, represented as fii, the average self-coancestry among the Ni individuals, represented as si, and the average inbreeding coefficient, represented as Fi = 2si − 1. Wright’s F-statistics [9] (also called Wright’s inbreeding coefficients) are calculated using the following formulae: where $F_{IS}=\frac{\tilde{F}-\bar{f}}{1-\bar{f}}$; $F_{ST}=\frac{\bar{f}-\tilde{f}}{1-\tilde{f}}$ and $F_{IT}=\frac{\tilde{F}-\tilde{f}}{1-\tilde{f}}$, where *F_IS_* is defined as the inbreeding coefficient of an individual with respect to the subpopulations, *F_ST_* is defined as the mean inbreeding coefficient of the subpopulation with respect to the entire metapopulation, *F_IT_* is defined as the inbreeding coefficient of an individual in relation to the entire population and while $\tilde{f}$ and $\tilde{F}$ are the average coancestry coefficient and inbreeding coefficient for the entire metapopulation and $\bar{f}$ is the average coancestry coefficient for the subpopulation. Wright’s inbreeding coefficients are not independent, as they are functionally interrelated since (1 − *F_IT_*) = (1 − *F_IS_*) (1 − *F_ST_*). The concept of population structure has more often been theorised as a deviation from the Hardy–Weinberg equilibrium, where *F_ST_*, for example, was originally defined as the correlation between random gametes within subdivisions (subpopulations) relative to gametes in the entire population. For the link between the original concept and Wright’s F coefficients estimated from pedigree, see Wright (1965) [12]. In population genetics theory, the *F_ST_* is often regarded as a parameter that quantifies genetic drift and is therefore interpreted as a genetic distance ranging between zero and one (no difference in allele frequency if the *F_ST_* is zero, or different alleles are fixed in each population if the *F_ST_* is one). In this concept, the population structure can be represented by an *F_ST_* matrix formed from the pairwise distances or the pairwise *F_ST_* coefficients (distances) between all subpopulations. Here, the *F_ST_* matrix was visualised using a heat map. The normality of the *F_ST_* coefficients was evaluated by an Anderson–Darling normality test. We checked all *F_ST_* values for every herd from which the largest value for each herd was taken. Then, we sorted these (maximum) *F_ST_* values from lowest to highest and calculated the proportion of the total population they represent. Finally, the results were depicted as histograms and the cumulative population proportion according to the maximum *F_ST_* values per each herd. For a comprehensive investigation of the relationships and distances between herds, we performed a multidimensional scaling analysis (MDS) [13] based on the pairwise *F_ST_* coefficients between herds, all of which were obtained from the *F_ST_* matrix.

### 2.3. Migration Assessment

The actual migration of pigs among herds derived from the stud book was visualised by the chord diagram. Every individual’s herd ID was documented from birth to the most recent assessment. Those with incomplete ID information at either birth or the present assessment were excluded from the parameter analysis. Chord diagrams were employed to illustrate the transition of individuals from their birth herds to another current herd, with separate diagrams for males and females, as well as a combined one. However, to enhance clarity, only individual chord diagrams for male and female migrations were presented in this paper.

### 2.4. Programme Used

The ENDOG version 4.8 [11] software programme was used to calculate the *F_ST_* matrix based on the differentiation between the herds (pairwise *F_ST_* coefficients). The subpopulations submenu in the population menu was used to compute F_ST_ values. The outcomes were documented in the table *Fis_Fsts* of a Gener.mdb file.

Based on the *F_ST_* results from ENDOG’s running, the *F_ST_* diagonal full matrices were created in Ms Excel 365. The function heatmap.2 of the R package “gplots” was used to create a heatmap to visualise the pairwise *F_ST_* matrix.

The chordDiagram function from the R package “circlize” [14] was used to create the chord diagram of migration intensity. The direction of arrows in the chord chart showed the direction of migration, and arrows sizes reflect the number of migrants. In addition, the Cmdscale function in R was used to perform the classical (metric) multidimensional scaling (MDS), also known as principal coordinates analysis [15].

## 3. Results

### 3.1. Population Subdivision

The population differentiation of the Blonde, Swallow-Belly and Red breeds is illustrated by the pairwise *F_ST_* coefficients, which are visualised in heatmap presentations (Figure 1, Figure 2, Figure 3 and Appendix A). The distribution of the *F_ST_* coefficients was not normal in any breed (*p* < 0.001). The average *F_ST_* coefficients were 0.04 for the Blonde and 0.047 for the Swallow-Belly and the Red, which are significant smaller than 0.05 (*p*-value < 0.05), while these parameters were even smaller (0.03 and 0.04, respectively) for the active herds. The heatmaps and histogram revealed that the Swallow-Belly breed has the highest prevalence of stratification herds (*F_ST_* > 0.15), followed by the Red and Blonde breeds (Appendix A, respectively). The proportion of herds with *F_ST_* > 0.15 was 15.96%, 12.41% and 12.40% respectively. In addition, the proportion of animals with an *F_ST_* bigger than 0.15 was 1.21%, 0.81% and 0.38%, respectively (Appendix A). In the currently active herds, highly differentiated herds with large distances (*F_ST_* > 0.15) were only observed in the Blonde and Red breeds, accounting for 6.41% and 3.64% (Figure 1, Figure 3, Figure 4a and Figure 5a), which represents a significant reduction compared to the total herds. A very small proportion of animal with an *F_ST_* bigger than 0.15 was found in the Blonde active herds, with 0.14%, and in the Red one, with 0.09% (Figure 4b and Figure 5b).

Within the Blonde Mangalica, three active herds (1645, 1630 and 1358) show considerable differentiation from each other (Appendix A). Despite the presence of these widely separated herds, the proportions of animals with *F_ST_* values bigger than 0.15 were 0.38% and 0.14% in the overall herds and in the active herds, respectively (Appendix A and Figure 4b). In addition, no visual groups were detected either in the overall herds (Appendix A) or in the active herds (Figure 6).

In the Swallow-Belly breed, a large differentiation was observed in herds 800, 1336, 1159 and 1494 (Appendix A). However, all active herds in this population had an *F_ST_* < 0.15, indicating that there was no significant genetic differentiation between the herds (Figure 2 and Figure 7a,b). In Figure 8, the active herds are scattered, but no sufficient groups were formed.

In the Red breed, large distances are observed between herds 198, 1436, 1646, 1385, 1325 and 1493 (Appendix A). The active herds in this breed showed a significant differentiation between herds 1436 and 1646 (Figure 3 and Figure 9). The *F_ST_s* remain consistent within the breed, and no visual groups are formed (Figure 9 and Appendix A). The proportions of animals with a maximum *F_ST_* by 0.15 were 0.81% and 0.09% in the overall herds and in the active herds, respectively (Figure 5b and Appendix A).

The three Mangalica breeds show a uniform *F_ST_* pattern between the herds. The small *F_ST_* group (*F_ST_* < 0.05) accounts for the largest proportion of more than 58% of the total, namely, 71.26%, 61.29% and 58.83% for the Blonde, Swallow-Belly and Red breeds, respectively (Table 2). Conversely, the large F_ST_ set (*F_ST_* > 0.15) represents a consistently minimal percentage of around 1.00%. The Red breed has the highest percentage (40.33%) of moderate differentiation between herds (0.05 < *F_ST_* < 0.15), followed by the Swallow-Belly breed with 37.27% and the Blonde breed with 27.55%. It is noteworthy that the Red breed shows a tendency to separate herds, with the majority of moderately differentiated herds. However, the strong stratification between herds was found in a very small proportion of only 0.84%.

While the proportion of the strongly differentiated herds was below 2.00% for the three breeds, the Blonde breed stands out with the highest average distance value (average *F_ST_*) within this group, which is 0.24 (between 0.15 and 0.35). In comparison, the Swallow-Belly and Red breeds are smaller, with an average *F_ST_* of 0.20 (between 0.15 and 0.35) and 0.21 (between 0.15 and 0.34), respectively. Conversely, the average *F_ST_* values for the small and moderate groups were approximately 0.03 and 0.07, respectively (Table 2).

Of the total herds, approximately 30% were active, with proportions of 30.23%, 32.98% and 35.71% for the Blonde, Swallow-Belly and Red breeds, respectively. Looking at the genetic distances between active herds, over 99.70% fall into the small and medium *F_ST_* groups. This leads to a remarkable decrease in the proportion of large *F_ST_* groups, which account for less than 0.30% in all three breeds, except for the Swallow-Belly breed, where the percentage is 0% (Table 2).

### 3.2. The Migration Assessment

The migration of individuals within herds was found to be considerable, affecting more than 60% of the total current herds. Specifically, this affected 61.63% of the Blonde breed, 75.53% of the Swallow-Belly breed and 63.64% of the Red breed. A consistent pattern emerged across all three breeds, suggesting that a significant number of females were transferred between herds, while in comparison, the movement of males remained relatively low (Appendix A and Figure 10, Figure 11, Figure 12). Within the three breeds, the herd numbered 872 was the most active and dominant in providing sires to neighbouring herds (Appendix A, Figure 10a, Figure 11a and Figure 12a).

In the Blonde breed, the maximum number of male pigs migrating from a particular farm to a particular herd was 10, exceeding the numbers for the Swallow-Belly and Red breeds, at 6 and 4, respectively. In contrast, the range for female pigs is much wider, reaching up to around 270 animals. In contrast the numbers for the Swallow-Belly and Red breeds were lower, with 86 and 78 individuals, respectively.

The connectivity among migrating herds revealed that more than 80% were connected by a single sire. More specifically, this percentage was 80.72% in the current Blonde herds, 87.00% in the current Swallow-Belly herds and 90.34% in the current Red herds. At the same time, 90% of these herds established connections involving more than two sows, and this pattern applied to all three breeds.

Within the Blonde breed, herd 872 stands out as the main source of sires for neighbouring herds, while herd 954 attracts the most out-migrating sires, as shown in Appendix A and Figure 10a. In addition, herds 1509 and 1466 play an important role in the migration of approximately 270 sows, as highlighted in Appendix A and Figure 10b.

Appendix A and Figure 11a show that herd 872 has the most significant migration, both in terms of out-migrating and incoming sires of the Swallow-Belly breed. In terms of the out-migration of females, there was significant movement from herd 721 to herd 1322, but the most significant influx was observed in herd 1460, as shown in Appendix A and Figure 11b.

For the Red breed, Appendix A and Figure 12a show that herd 872 gave most sires to neighbouring herds, while herd 751 received most sires and shared them with other breeds. For female pigs, herd 675 was the largest contributor and herd 657 is the largest recipient, as shown in Appendix A and Figure 12b.

## 4. Discussion

Some studies on genetic variability between breeds have been carried out in Mangalica pigs [6,16]. A study on the Hungarian population of Mangalica pigs genotyped at 10 microsatellite loci revealed the presence of three clusters representative of three different breeds, namely, Swallow-Belly, Red and Blonde [6]. However, analyses utilising mtDNA markers were unable to distinguish subpopulations within this Mangalica population [16]. Although studies using different methods do not consistently separate the three different breeds, the three different coat colour variants of the Mangalica in Hungary are treated as if they were three separate breeds in the context of breeding management and breed conservation. There is no crossbreeding between these different breeds. A study of the genetic variability within the populations and structure of these breeds could shed light on their evolutionary patterns during more than four decades of conservation efforts in numerous Hungarian herds.

Traditionally, conservation efforts have focused on diversity between breeds, because according to Barker [17], the most important goal in conserving the diversity of domestic animals is the conservation of specific breeds. However, it is argued that approaches that emphasise the component of genetic diversity between breeds may not be the most effective, as they neglect the component of variation within breeds [18,19,20]. According to Cervantes et al. [7], accessing genetic variability within populations, understanding population structures and analysing gene flow are crucial steps in the implementation of selection programmes. This assessment plays a central role in the formulation of efficient management strategies for genetic stock with the aim of improving the genetic basis for selection purposes. According to Molnár et al. [16], populations within a breed that are geographically and/or ecologically isolated may acquire different physiological traits due to the specific selection criteria applied in the breeding process. Consequently, these isolated populations may differ genetically from other populations of the same breed that exhibit similar phenotypes, which may result in them being recognised as different breeds [16]. According to Wilkinson et al. [21], the genetic substructure within a breed, as revealed by individual clustering methods, is likely to be rare in domestic species, with the presence of a limited genetic substructure typically observed in only one or two exceptional breeds. However, within-breed stratification has been detected in several livestock species, e.g., chickens [21], horses [22], castles [23], goats [24,25], rabbits [26,27], dogs [28,29] and pigs [30,31].

The estimated *F_ST_* coefficients provide information on the degree of differentiation between a group of populations, as applied in the present study to assess the differentiation between the herds of three Mangalica breeds. The *F_ST_* coefficients, which range from 0 to 1, indicate the extent of genetic differentiation. A value of zero means that the genetic material is completely shared, allowing free crossbreeding. In contrast, a value of one indicates that all genetic variation is integrated into the population structure, meaning that there is no shared genetic divergence and populations are considered fixed or divergent [32]. The interpretation guidelines for Wright’s *F_ST_* coefficient were presented by Hartl and Clark [32] as follows: an *F_ST_* below 0.05 indicates low genetic differentiation; an *F_ST_* from 0.05 to 0.15 indicates moderate genetic differentiation; an *F_ST_* from 0.15 to 0.25 indicates  high genetic differentiation; an *F_ST_* above 0.25 indicates very high genetic differentiation. In addition, Frankham et al. [33] reported that *F_ST_* values above 0.15 indicate significant differentiation, while *F_ST_* values below 0.05 indicate insignificant differentiation. In the present study, the *F_ST_* coefficients between herds were between 0.00 and 0.35, as indicated by the colour spectra in the heatmap colours (Appendix A). Most of the analysed herds, representing more than 58% of the total population (Table 2), showed insignificant genetic differentiation according to the guidelines of Frankham et al. (2002) [33]. This group is even more dominant in active herds, with more than 65%. The *F_ST_* range estimated in this study is broader compared to that in the research on Greek black pigs [34], where the *F_ST_* values range from 0.058 to 0.291. Similarly, it exceeds the *F_ST_* reported in studies on four commercial pig breeds and on Monteiro pigs, with *F_ST_* values ranging between 0.067 and 0.168 [30] and from 0.009 to 0.063 [35], respectively. The variations observed may be attributed, in part, to the fact that the current study estimates *F_ST_* based on pedigree data, as opposed to the molecular data used in other research. Although the *F_ST_* range in this study is quite broad, with some herds reaching up to 0.35, the average *F_ST_* coefficient across herds is not high (less than 0.047), which is significantly lower than 0.05 (*p*-value < 0.05). This *F_ST_* value is higher than the *F_ST_* of 0.006 observed in the study of black Slovenian pigs [36] but lower than the inter-herd *F_ST_* (*F_ST_* = 0.065) within the same commercial breeds according to Snegin et al. (2021) [30]. It means that there are some divergent herds, but overall, there is a low genetic differentiation between herds in the three breeds analysed.

In addition, multidimensional scaling showed that the populations analysed were insufficient in forming groups for dimensions 1 and 2 (Appendix A, Figure 6, Figure 8 and Figure 9), although there are some visually divergent herds. Among the active herds, the Swallow-Belly pigs showed quite a large differentiation from each other, but also without a defined structure (Figure 8). This could be due to the small population size of this breed. In pigs, Wilkinson et al. [31] used a Bayesian analysis of population structure based on genotypic data to detect a substructure within the British Meishan breed, but this was not present in other methods. Snegin et al. [30] found high variability between individual herds within the four commercial pig breeds, which contributed to the significant differences between the breeds analysed.

The Swallow-Belly and Red breeds showed a stronger tendency towards internal differentiation, with a larger percentage of herds showing moderate genetic differences than in the Blonde breed. Nevertheless, the average *F_ST_* coefficients between herds remained similar for all three breeds (0.04). This phenomenon may be explained by the smaller population sizes of the Swallow-Belly and Red breeds.

Genetic differentiation was observed in certain herds across the entire populations analysed (Appendix A), but these were unable to form a substructure (Appendix A). The populations analysed, which have been listed in pedigrees since 1981, include both active and previously inactive herds. Analysing entire populations provides a comprehensive overview, but precise information on genetic subdivision depends on the active herds. Among the active herds, these differentiated herds make up a tiny proportion, less than 0.30% (Table 2). When analysing these herds, e.g., 1645 and 1630 in the Blonde breed (Figure 6) and 1436 and 1646 in the Red breed (Figure 9), each herd had only one selected sire. When calculating the average herd coancestry, the predominance of self-coancestry contributes to high *F_ST_* coefficients. Consequently, this leads to a clear separation from other groups, as shown in Figure 6 and Figure 9. However, despite this observed differentiation, the details of the substructure within the herds remain unclear in the overall view.

The results showed a strong migration between the herds of the three breeds, as about 60% of the herds are connected to other herds in some way. In addition, more than 90% of the migration involved one sire and more than two sows. The extensive exchange of animals between individual herds could be the reason for genetic similarity between herds in this study. Achmann et al. [37] found in the studies of Lipizzaners that the exchange of horses between studs plays a crucial role in mitigating genetic divergence between subpopulations. Dumasy et al. [38] found that an increase in genetic distance was due to a reduced connectivity between herds. This conclusion was drawn by examining the correlation between Reynolds genetic distances and the shortest path lengths calculated by the exchange network method. In addition, the Blonde breed has a lower average *F_ST_* (0.04) compared to the Swallow-Belly and Red breeds (0.047), which could be due to a higher number of exchanged animals between herds within the Blonde breed. Dumasy et al. [38] pointed out the importance of considering the number of exchanged animals when explaining genetic differentiation, and the increase in exchanged animals within the Blonde breed was consistent with its lower *F_ST_* coefficient. Both male and female individuals play crucial roles in creating robust connections between herds within breeds. Significantly more females were exchanged in the breeds analysed. However, it must be noted that the two sexes show different migration characteristics. The exchange of boars between herds is a continuous process, and generally, it consists of one or few animals. On the contrary, the female exchange is occasional, and its aim can be establishing a new herd, the herd size enlargement of an existing herd or the re-establishment of a previously closed herd. In addition, the Mangalica farms do not use artificial insemination (personal communication with Hungarian National Association of Mangalica Breeders), but the boars are moved between herds under control. This could support the contribution to the gene flow between the herds of sires.

According to Snegin et al. [30], the differentiation within the breed has been attributed to many factors, including the gene flow, geographic isolation, breeding preferences and distinctive genetic backgrounds found in the genealogical groups (sire/dam lines) of the breed’s founders. Geographic isolation contributed to the emergence of intra-breed differentiation in local goats in Spain and Portugal [24]. However, there was no impact of geographical differences on genetic differentiation within the Monteiro pig breed in the Brazilian Pantanal Ecosystem. This is attributed to the presence of evidence indicating a high level of gene flow within this population [35]. This can be linked to the present study; even Mangalica pigs are kept in many different regional herds, and the intensive connection between herds has contributed to the low genetic differentiation. Differentiation in dog breeds, as demonstrated by Wiener et al. [29], was driven by the direction of breeding or artificial selection [29]. This is primarily not happening in the current study, as all registered herds follow the same breeding strategy prescribed by the Hungarian National Association of Mangalica Breeders. In addition, no significant barriers to gene exchange were identified in this study.

## 5. Conclusions

Utilising the multidimensional scaling and visualising of Wright’s *F_ST_* coefficients, the substructure within the Blonde, Swallow-Belly and Red breeds cannot be delineated. Furthermore, the frequency of extensive animal exchange between individual herds, the uniformity of mating strategies and the lack of significant barriers to gene exchange confirm genetic homogeneity within these breeds. The patterns observed indicate that the breeds studied, with the aim of maintaining genetic diversity and minimising the risk of inbreeding, show positive signs consistent with conservation objectives.

## Figures and Tables

**Figure 1 animals-14-00653-f001:**
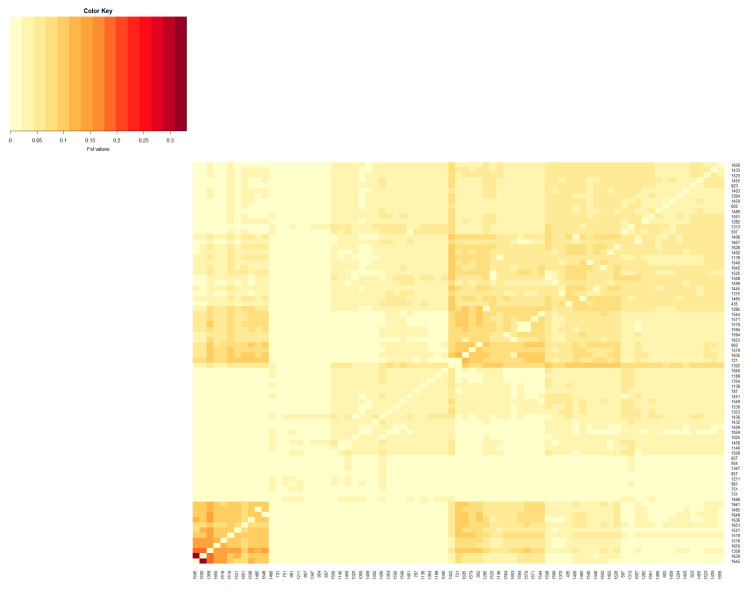
Heatmap based on pairwise *F_ST_* coefficients between the active herds of the Blonde Mangalica breed. The color key legend ranging from yellowish to dark red represents the scale of *F_ST_* coefficients from zero onward. The *x* and *y* axes show the herd name.

**Figure 2 animals-14-00653-f002:**
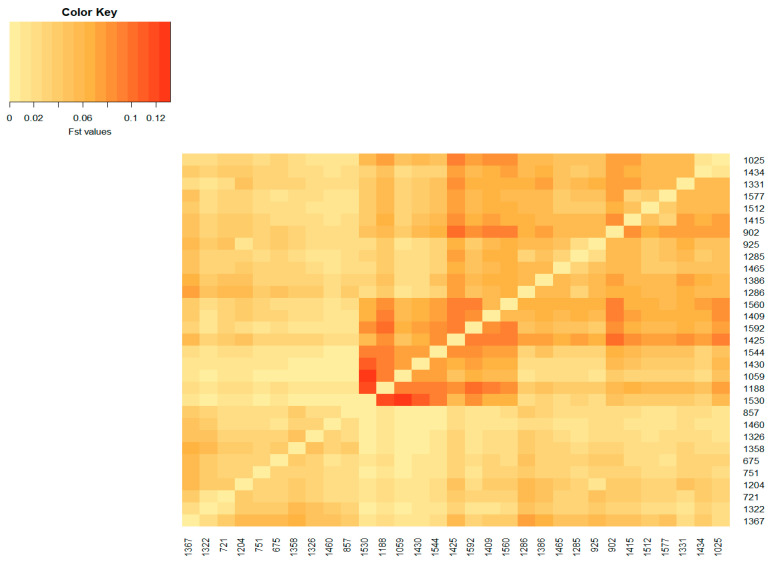
Heatmap based on pairwise *F_ST_* coefficients between the active herds of the Swallow_Belly Mangalica breed. The color key legend ranging from yellowish to dark red represents the scale of *F_ST_* coefficients from zero onward. The *x* and *y* axes show the herd name.

**Figure 3 animals-14-00653-f003:**
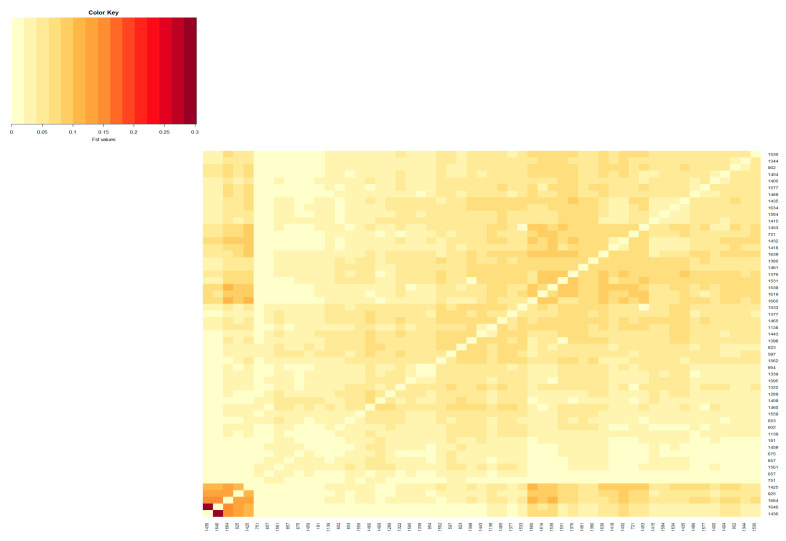
Heatmap based on pairwise *F_ST_* coefficients between the active herds of the Red Mangalica breed. The color key legend ranging from yellowish to dark red represents the scale of *F_ST_* coefficients from zero onward. The *x* and *y* axes show the herd name.

**Figure 4 animals-14-00653-f004:**
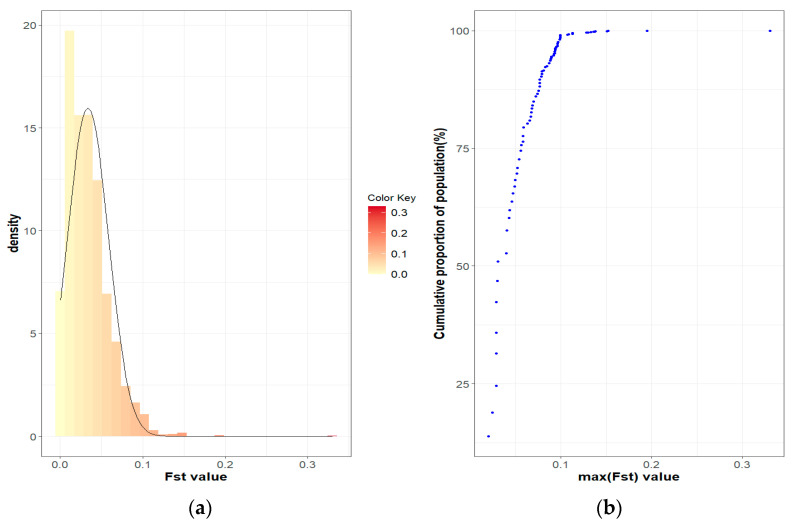
*F_ST_* coefficients in the Blonde Mangalica active herds: (**a**) Histogram of *F_ST_* values with density; (**b**) Cumulative proportion of the population related to the maximum *F_ST_* of the herds.

**Figure 5 animals-14-00653-f005:**
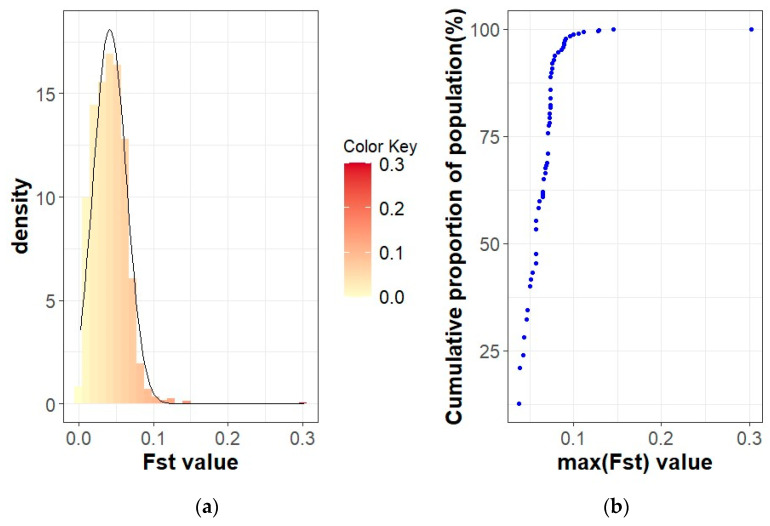
*F_ST_* coefficients in the Red Mangalica active herd: (**a**) Histogram of *F_ST_* values with density; (**b**) Cumulative proportion of the population related to the maximum *F_ST_* of the herds.

**Figure 6 animals-14-00653-f006:**
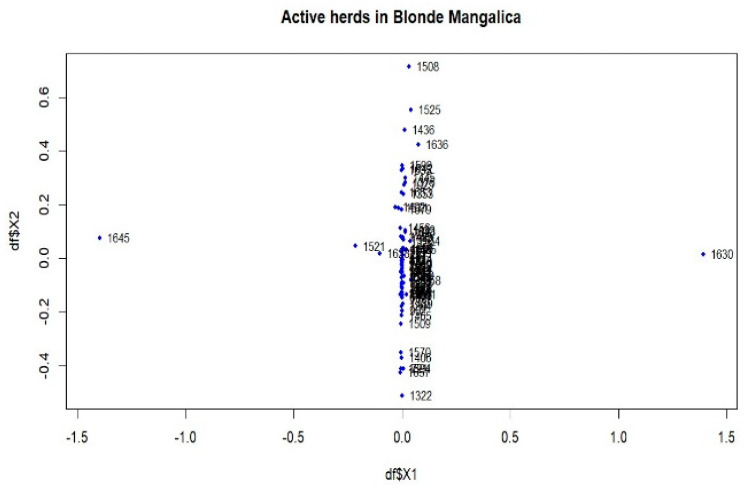
Multidimensional scaling (MDS) plot (MDS1&MDS2) of active herds of the Blonde Mangalica.

**Figure 7 animals-14-00653-f007:**
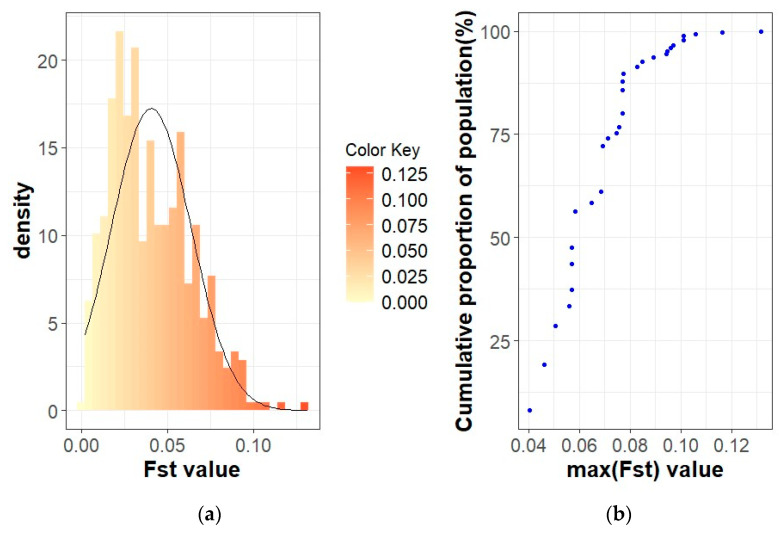
*F_ST_* coefficients in the Swallow-Belly Mangalica active herds: (**a**) Histogram of *F_ST_* values with density; (**b**) Cumulative proportion of the population related to the maximum *F_ST_* of the herds.

**Figure 8 animals-14-00653-f008:**
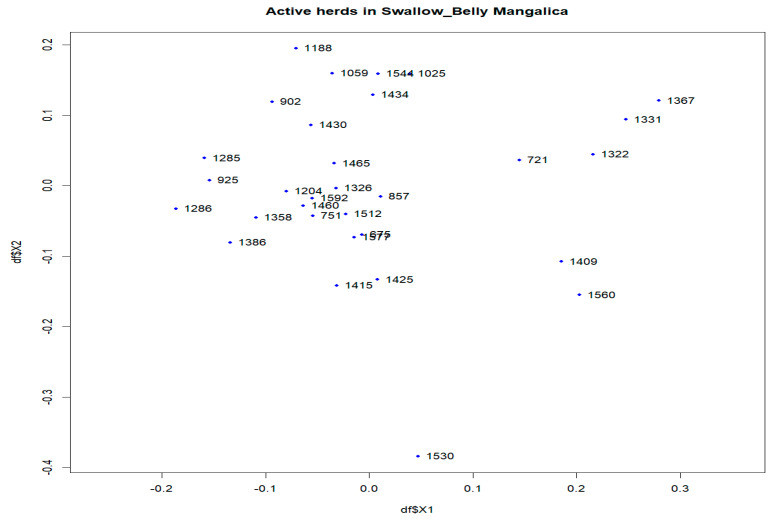
Multidimensional scaling (MDS) plot (MDS1&MDS2) of active herds of the Swallow-Belly Mangalica.

**Figure 9 animals-14-00653-f009:**
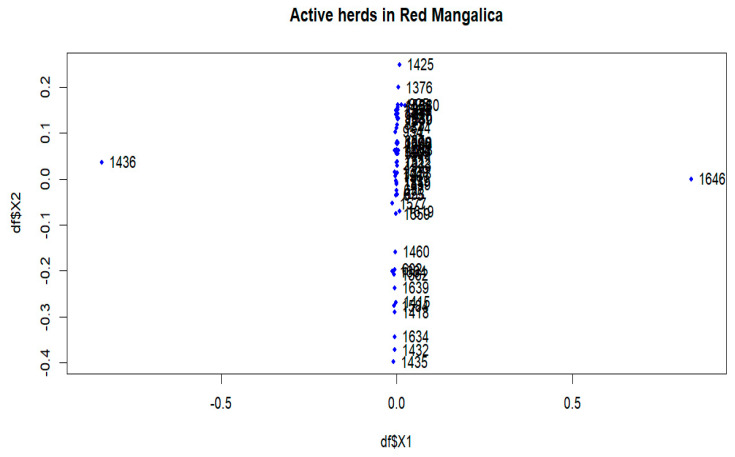
Multidimensional scaling (MDS) plot (MDS1&MDS2) of active herds of the Red Mangalica.

**Figure 10 animals-14-00653-f010:**
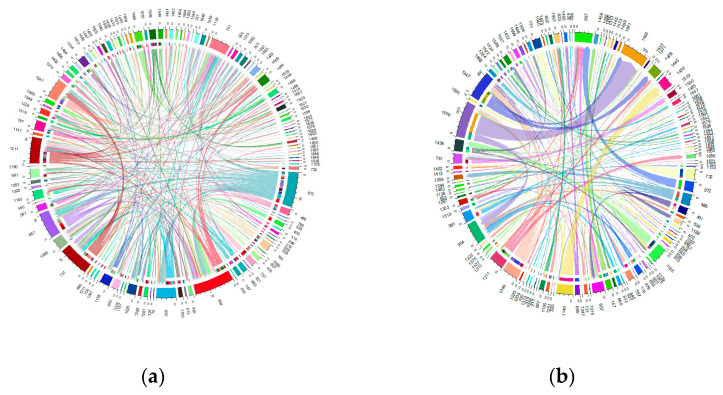
Migration of the Blonde breed in active herds: (**a**) Male; (**b**) Female.

**Figure 11 animals-14-00653-f011:**
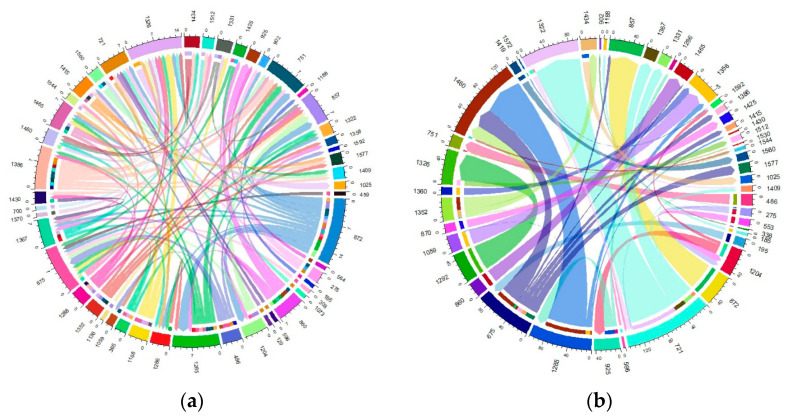
Migration of the Swallow-Belly breed in active herds: (**a**) Male; (**b**) Female.

**Figure 12 animals-14-00653-f012:**
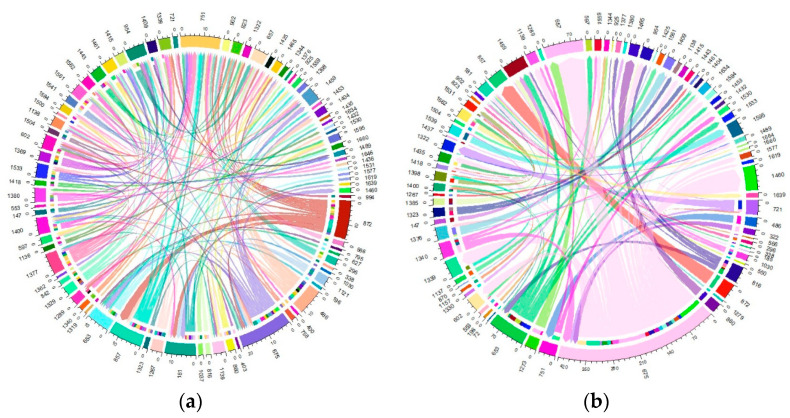
Migration of the Red breed in active herds: (**a**) Male; (**b**) Female.

**Table 1 animals-14-00653-t001:** Number of herds and breeding animals of three-colour variations of Mangalica pigs.

	Blonde	Swallow-Belly	Red
Total number of individuals	14,550	2638	4566
Total number of herds	258	94	145
Number of active herds	78	31	55
Total number of sires	748	237	305
Number of active sires	427	129	305
Total number of sows	6393	1094	1779
Number of active sows	3944	669	1779

**Table 2 animals-14-00653-t002:** Average pairwise F_ST_ coefficients among herds sorted by differentiation intensity.

Breed	*F_ST_*_Group	Total Herds			Active Herds		
		N	Mean	Percent	N	Mean	Percent
Blonde	S	23,625	0.02 ± 0.014	71.26	2368	0.02 ± 0.014	78.85
	M	9133	0.07 ± 0.019	27.55	627	0.07 ± 0.017	20.88
	L	395	0.24 ± 0.077	1.19	8	0.18 ± 0.062	0.27
Swallow-Belly	S	2679	0.03 ± 0.013	61.29	310	0.03 ± 0.012	66.67
	M	1629	0.07 ± 0.020	37.27	155	0.07 ± 0.014	33.33
	L	63	0.20 ± 0.052	1.44	0	0	0
Red	S	6142	0.03 ± 0.013	58.83	972	0.03 ± 0.013	65.45
	M	4210	0.07 ± 0.018	40.33	512	0.06 ± 0.013	34.48
	L	88	0.21 ± 0.058	0.84	1	0.30	0.07

*F_ST_*: Wright’s *F_ST_* coefficient, S: *F_ST_* ≤ 0.05, M: 0.05 < *F_ST_* ≤ 0.15, L: *F_ST_* > 0.15, N: number of observations.

## Data Availability

Restrictions apply to the availability of these data. Data were obtained from the Hungarian National Association of Mangalica Breeders and are available from the corresponding author with the permission of the Hungarian National Association of Mangalica Breeders.

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
