# Peer review of "Population Subdivision and Migration Assessment of Mangalica Pig Breeds Based on Pedigree Analysis"

_animals, 2024, doi:10.3390/ani14040653_

Round 1
Reviewer 1 Report
Comments and Suggestions for Authors
This study investigated Mangalica pig breeds based on pedigree analysis, addressing an under-researched area of population subdivision and migration intensity. The study focus on a specific breed with distinct color variations (Blonde, Red, Swallow Belly), and offers a contribution to the field of genetic and conservation studies. The manuscript provides a valuable contribution to the understanding of genetic diversity and population dynamics in Mangalica pig breeds. However, expanding its comparative context and addressing some minor grammatical issues would further enhance its quality.
The main results (figures) of the study are in the Supplementary Materials, which is not conducive to readers' reading. It is suggested to select some representative figures and put them in the Results.
Line 75-92: Please use a table to show the information of each sub-population.
Line 118: The method of visualization of the chord diagram Migration density should be described in details. Only one sentence, it’s so simple.
Line 120-126: The related operations and parameters of using these softwares need to be briefly explained.
Line 177: “FST: fixation index”, But in the previous materials and methods, this Fst refers to: “pairwise genetic distances (Fst)” (line 121) or “the mean inbreeding coefficient of subpopulation relative to the entire population” (line 108). Please unify the interpretation of this indicator Fst in the full text. It should be a unique indicator.
Line 243: “Estimating the Fixation Index (FST) provides insights into the degree of”. What’s Fixation Index (FST)?
Comments on the Quality of English LanguagePlease check grammatical errors carefully.
for example:
1. Line 73: "since all the breeds have been being kept" change to "since all the breeds have been kept".
2. Line 100: "between-breed diversity may be not optimal" should be "may not be optimal".
3. Line 219: "Blonde Swallow-Belly and Red Mangalica breeding animals (i.e., that had produced offspring)." change to "Blonde, Swallow-Belly, and Red Mangalica breeding animals, i.e., those that had produced offspring."
4. Line 350: "ENDOG software programs" should be "ENDOG software program".
5. Line 412: "large distances were calculated in herds 800 1336 and 1159" - the list needs commas for clarity.
6. Line 723: "Nevertheless the average FST values between" - Consider adding a comma after "Nevertheless,".
Author Response
Reviewer 1
Issue 1: The main results (figures) of the study are in the Supplementary Materials, which is not conducive to readers' reading. It is suggested to select some representative figures and put them in the Results.
Answer: Thank you for your suggestion. Figures of the current active herds of Swallow-belly Mangalica are moved to the main text and reodered
Issue 2: Line 75-92 please use a table to show the information of each sub-population.
Answer: Thank you for your recommendation. The suggested table is included.
Issue 3: Line 118- the method of visualization of the chord diagram migration density should be described in detail. Only one sentence, it’s so simple.
Answer: The actual migration intensity of pigs among herds derived from the stud book was visualized by the chord diagram. Every individual's herd ID was documented from birth to the most recent assessment. Those with incomplete ID information at either birth or the present assessment were excluded from the parameter analysis. Chord diagrams were employed to illustrate the transition of individuals from their birth herd to another, with separate diagrams for males and females, as well as a combined one. However, to enhance clarity, only individual chord diagrams for male and female migrations were presented in this paper.
Issue 4: Line 120-126_ The related operations and parameters of using these softwares need to be briefly explained.
Answer: The ENDOG version 4.8 [13] software programme was used to calculate the FST matrix based on the differentiation between the herds (pairwise FST coefficients). The subpopulations submenu in the population menu was used to compute FST values. The outcomes were documented in the table Fis_Fsts of Gener.mdb file.
Based on the FST results from ENDOG’s running, the FST diagonal full matrices were created in Ms Excel 365. The function heatmap.2 of the R package "gplots" was used to create a heatmap to visualise the pairwise FST matrix.
The chordDiagram function from the R package "circlize" [14] was used to create the chord diagram of migration intensity. The direction of arrows in the chord chart showed the direction of migration and arrows sizes reflect number of migrants. In addition, the Cmdscale function in R was used to perform the classical (metric) multidimensional scaling (MDS) also known as principal coordinates analysis [15].
Issue 5: Line 177_ “FST: fixation index”, But in the previous materials and methods, this Fst refers to: “pairwise genetic distances (Fst)” (line 121) or “the mean inbreeding coefficient of subpopulation relative to the entire population” (line 108). Please unify the interpretation of this indicator Fst in the full text. It should be a unique indicator.
Answer: authors can confirm that FST is a unique indicator. It is part of the concept of Wrigth’s F-statistics where FST is a measure of genetic divergence among subpopulations.
Issue 6: Line 243_ “Estimating the Fixation Index (FST) provides insights into the degree of”. What’s Fixation Index (FST)?
Answer: FST is sometimes called “fixation index (Allendorf et al. 2022: Conservation and the Genomics of Populations, Third Edition, Oxford University Press, p. 174). In order to increase clarity, the term fixation index was removed from the manuscript.
Reviewer 2 Report
Comments and Suggestions for Authors
The authors assess the extent of the population subdivision of Mangalica pigs and the contribution of migration intensity to their substructure. And performed genealogy analysis for breedinganimals born between 1981 and 2023 examining three colour variations (Blonde, Swallow-Belly, ancRed). This research is very interesting, but there are still some obvious shortcomings, which need to be carefully revised and improved by the author.
1、Line 67-70,The authors claims that “all the breeds are kept in multiple herds these herds can be interpreted as subpopulations” is not sufficiently rigorous. A subpopulation, also known as a "geographical subspecies," refers to a group of individuals within a species that are geographically and ecologically isolated enough to exhibit distinct morphological, physiological, and genetic characteristics, including different geographical distributions and diverse ecological environments.
2、Please replace the Supplementary Figure 2, 5, and 8 with high-definition figures.
3、The article is descriptive and difficult to understand for readers in non-quantitative genetics fields. Authors are advised to add graphic abstracts to deepen readers' understanding of the content of the article
4、The author should include the Figures that support the main conclusions in the main text. For the Heatmap and dendrogram (Fig.S1, S4, S7), the author did not provide sufficient description of the data and differences presented in that section.
5、In the discussion section, the author did not effectively compare and analyze the research results with existing studies.
6、The author needs to re-describe the Figures in the article because some errors were discovered during the examination of the manuscript.
6.1 Line138-141,the author's description of Figure S1b and Figure S2b does not seem to correspond accurately to the figures. For example, in Figure S2b, the herds labeled as "1630" appears to be a distant cluster.
6.2 Line142-146,Within Swallow-Belly breed, according to Figure S4a,large distances are observed among 4 herds active herds (1645, 1630, 1358,and 1494), not the 3 active herds (1645, 1630, and 1358) described by the author.
6.3 Line 147-150, In the Red breed, for Figure S7a, S8a, S8b, It seems that herd 1646 was not found, and herd 1500 was absence.
7、Line 311-315, The authors claims that all registered herds adhere to the same breeding strategy mandated by the Hungarian National Association of Mangalica Breeders. And there are no notable barriers to gene exchange discovered during the investigation. It seems to conflict with the Result 3.2 (Across all three breeds…remained relatively minimal in comparison). The author should carefully analyze the results produced in the discussion section.
Author Response
Reviewer 2
Issue 1: Line 67-70,The authors claim that “all the breeds are kept in multiple herds these herds can be interpreted as subpopulations” is not sufficiently rigorous. A subpopulation, also known as a "geographical subspecies," refers to a group of individuals within a species that are geographically and ecologically isolated enough to exhibit distinct morphological, physiological, and genetic characteristics, including different geographical distributions and diverse ecological environments.
Answer: We agree with your definition of subpopulation especially in the context of wild species. Even if the herds of the studied populations may not rigorously satisfy the subpopulation definition, we wanted to analyze if keeping in multiple herds for more than 40 years caused stratification that has not yet been discovered.
Issue 2: Please replace the Supplementary Figure 2, 5, and 8 with high-definition figures.
Answer: According to your request, the mentioned figures were replaced
Issue 3: The article is descriptive and difficult to understand for readers in non-quantitative genetics fields. Authors are advised to add graphic abstracts to deepen readers' understanding of the content of the article.
Answer: We fully agree with the remark that in general any manuscript should be prepared in a way that even non-specialists should be able to understand it. However, according to our opinion the wording of the text cannot be further simplified in order not to decrease scientific quality. We are still hoping that readers from the animal breeding field would be able to understand our work.
Issue 4: The author should include the Figures that support the main conclusions in the main text. For the Heatmap and dendrogram (Fig.S1, S4, S7), the author did not provide sufficient description of the data and differences presented in that section.
Answer: Figures of the current active herds of Swallow-belly Mangalica are moved to the main text and reordered. There are some adjustments with the heatmaps that the dendrograms are removed and histogram are added.
Issue 5: In the discussion section, the author did not effectively compare and analyze the research results with existing studies.
Answer: Five more research papers [11], [15], [34], [35],[36] related to pig are referred to compare and analyze the research.
Issue 6: The author needs to re-describe the Figures in the article because some errors were discovered during the examination of the manuscript.
6.1 Line138-141,the author's description of Figure S1b and Figure S2b does not seem to correspond accurately to the figures. For example, in Figure S2b, the herds labeled as "1630" appears to be a distant cluster.
Answer: we have checked and preserve our description. For example, in Figure S2b, many herds are scatted from the center looking like distant herds, but these does not show clusters.
6.2 Line142-146,Within Swallow-Belly breed, according to Figure S4a,large distances are observed among 4 herds active herds (1645, 1630, 1358,and 1494), not the 3 active herds (1645, 1630, and 1358) described by the author.
Answer: There may be a misunderstanding here. In our manuscript, Line 142 – 146 described Swallow-Belly Mangalica: “Concerning the Swallow-Belly breed, large distances were calculated in herds 800, 1336, and 1159 (Figures S5, S7 is replaced). However, all active herds in this population showed FST < 0.15, indicating the absence of substantial genetic differentiation among the herds (Figure S4b). The active herds are scattered in Figures S5c and S5d, but clusters are not sufficiently formed.”. So, the herd 1494 is added in this paragraph.
6.3 Line 147-150, In the Red breed, for Figure S7a, S8a, S8b, It seems that herd 1646 was not found, and herd 1500 was absence.
Answer: Figure S7a did not present all herd ID because the figure was magnified, and some herd ID were out of the figure. A new Figure S9a is replaced with figure S7a, and the Figure shows that herd named 198, 1385, 1325, 1493, 1646, and 1436 are greatly different with each other.
Issue 7: Line 311-315, The authors claims that all registered herds adhere to the same breeding strategy mandated by the Hungarian National Association of Mangalica Breeders. And there are no notable barriers to gene exchange discovered during the investigation. It seems to conflict with the Result 3.2 (Across all three breeds…remained relatively minimal in comparison). The author should carefully analyze the results produced in the discussion section.
Answer: The related part of the discussion was changed in order to clarify the results.
Reviewer 3 Report
Comments and Suggestions for Authors
Hungarian Mangalica are bred within three different colour variants including Blonde, Red, Swallow Belly since 1976. The aim of this study to evaluate possible subdivision of the population structure. Wright F-statistics were used to reach this objective. For the analysed breeds not any signs of sub-structure could be observed.
Abstract: may be expanded to explain why substructure may be a problem in breed conservation. Also some outlook on further work in conservation might be useful in order to see whether the endangerment levels of these breeds.
Introduction: some more explanations and outlines to the development and endangermant stauts may be desired to give mor insights in these population dynamics of these breeds.
Lines 94-116: formulas are not well explained.
Line 111-112: 𝑓 is the average coancestry for the subpopulation.
You have 3 subpopulations. It is not clear how do you deal with these 3 subpulation using the formulas given.
It may be better to explain that all these formulas refer to pairwise comparisons. Then the question arises, do have several metapopulations or just one comprising all 3 breeds.
Line 118: please define migration intensity and show how you did calculate this parameter. Are you distinguishing females and males in the migration rate. Is artificial insemination used in these breeds and may AI centribute to migration intensity. Particularly, in case of AI, this issue has to be outlined and regarded in data analysis.
There are many supplementary figures which need more explanations.
There are no statistics if significant subclusters were detected. The evaluation of the clusters relies on visual judging of the figures.
Some subclusters may be expected due to the herd structure and the number of litters produce per sow in her lifetime. In case of many long living sows, substructure may be favored.
Substructure may be lower if the number of litters per sow und lifetime per sow are shorter. So, there may be some competing goals. Can you discuss this issue. Similar in boars. Are there restriction is use of boars per sow.
The discussion concentrates on previous reports and generally on the approach with FST. Discussion of the specific situation is too short and should be expanded.
My advice for the authors is to use statistical significance tests to validate splitting in subclusters. There are procedures and software available to do this. This would be a worthwile piece for this study.
Comments on the Quality of English LanguageNo comments
Author Response
Reviewer 3
Issue 1: Abstract may be expanded to explain why substructure may be a problem in breed conservation. Also, some outlook on further work in conservation might be useful in order to see whether the endangerment levels of these breeds.
Answer: the abstract and simple summary were modified
Issue 2: Introduction: some more explanations and outlines to the development and endangermant stauts may be desired to give more insights in these population dynamics of these breeds.
Answer: According to our opinion further information about these breeds can be obtained from our previous study (Nguyen et al. 2023: Livestock Science, 273, 105265).
Issue 3: Lines 94-116: formulas are not well explained.
Answer: This part was rewritten explaning the formulas
Issue 4: Line 111-112: f is the average coancestry for the subpopulation. You have 3 subpopulations. It is not clear how do you deal with these 3 subpopulations using the formulas given. It may be better to explain that all these formulas refer to pairwise comparisons. Then the question arises, do have several metapopulations or just one comprising all 3 breeds.
Answer: In this research, 3 color variations of Mangalica, namely Blonde, Swallow Belly, and Red were considered as three different populations. In which the herd IDs of every population were assumed as subpopulations.
Issue 5: Line 118- please define migration intensity and show how you did calculate this parameter. Are you distinguishing females and males in the migration rate. Is artificial insemination used in these breeds and may AI centribute to migration intensity. Particularly, in case of AI, this issue has to be outlined and regarded in data analysis.
Answer: In this part we describe the migration of pigs between herds by calculate the percentage of migrant in total population of breed. The migration of male and females are calculated both separation and combination, but for clear view, the separated calculation were presented. The farms do not use AI so that this issue was not discussed.
Issue 6: There are many supplementary figures which need more explanations.
Answer: This section has been modified.
Issue 7: There are no statistics if significant subclusters were detected. The evaluation of the clusters relies on visual judging of the figures. Some subclusters may be expected due to the herd structure and the number of litters produce per sow in her lifetime. In case of many long living sows, substructure may be favored. Substructure may be lower if the number of litters per sow und lifetime per sow are shorter. So, there may be some competing goals. Can you discuss this issue. Similar in boars. Are there restriction is use of boars per sow. My advice for the authors is to use statistical significance tests to validate splitting in subclusters. There are procedures and software available to do this. This would be a worthwile piece for this study.
Answer: We agree with the reviewer that this section has large importance. Actually it was not our intention to use cluster analysis it was only included in the default setting creating the heatmaps. Concerning MDS according to our knowledge interpreting the results is only made by visual inspection, evaluating if the herdIDs form several well distinguished groups and there is no statistical test determining the number of possible subpopulations. We would like to emphasize that our conclusion was most based on the fact that among the active herds practically all showed FST < 0.15 which means that there are not manifest significant divergence. The manuscript has been extended with histograms providing a very clear overview of the pairwise FST values among the herds.
Issue 8: The discussion concentrates on previous reports and generally on the approach with FST. Discussion of the specific situation is too short and should be expanded.
Answer: The discussion was expanded in comparison to other research [30], [34], [35], [36]
Round 2
Reviewer 1 Report
Comments and Suggestions for Authors
After authors’ revision, my concerns have been addressed. However, there are still several problems worthy of attention.
Line 79-80: Not needing approval and not getting approval are two different things.
Table 1: Please pay attention to the standard use of punctuation marks. Do they conform to the style or requirements of Animals? I don't know why there is a blank page at the end of this form.
Table 2: the same question as Table 1.
Figrue 4a, 4b: I didn't see Figure 3. Why did Figure 4 appear?
The main results of this paper are in the supplementary file, which will make it inconvenient for readers to read. The Journal Animals is not calculated the publication fee by the number of pages, and it is published electronically on the Internet. Therefore, it is better to put all the Figures in the text, not in the supplementary file.
Author Response
Review 1
1/ Line 79-80: Not needing approval and not getting approval are two different things.
Answer: Because the current study exclusively involved the analysis of genealogical data stored in datasets, approval from the Animal Care and Use Committee was not needed
2/ Table 1: Please pay attention to the standard use of punctuation marks. Do they conform to the style or requirements of Animals? I don't know why there is a blank page at the end of this form.
Answer: a comma was put in thousands. The blank was removed.
3/Table 2: the same question as Table 1
Answer: a comma is put in thousands. The blank is removed.
4/ Figrue 4a, 4b: I didn't see Figure 3. Why did Figure 4 appear?
Answer: Since the file was modified with track change, the figures and captions are sometimes not fixed in a right position. This problem was handled.
5/ The main results of this paper are in the supplementary file, which will make it inconvenient for readers to read. The Journal Animals is not calculated the publication fee by the number of pages, and it is published electronically on the Internet. Therefore, it is better to put all the Figures in the text, not in the supplementary file.
Answer: Thank you so much for your recommendation! We realize that following too many figures in the main text is also inconvenient for readers. Thus, all figures related to active herds were shown in the main text while the figures describing entire populations were placed in the supplementary.
Reviewer 3 Report
Comments and Suggestions for Authors
The authors have improved their manuscript and added additional explanations to improve the understanding of the outcomes of their study.
Comments:
Line 28: "and chord diagrams" , please delete.
Line 29: "significant" , please delete, because no statistical tests shown in Results.
Can you please show the average coancestry and inbreeding coefficients for the three Mangalica breeds and their development over birth years. Then you can test whether coancestry and inbreeding were constant or shows a significant trend.
In addition, you could show confidence intervals for F-ST values and look at Box-plots and qq-plots if normally distributed. This would also help to evaluate these values.
Line 35: genealogy, not really done.
Line 70-73: rather general; please explain why you conclude that you have a substructure. Farm animals are usually kept in herds. Also give some important results of the demographic analysis which may important for your present study.
Line 86: inTable 1. >> in Table 1.
Line 92: Table1 >> Table 1
Legend of Table 1 should describe the table contents. please amend.
Line 130 ff: please describe how did you calculate migration rate. Not clear.
Did you calculate number of immigrants minus number of emigrants in the study period divided by the total number of animals being kept in the respective herd with immigration over this period? Please clarify, Did you use weights by herd size?
Checking these rate between breeds and their distributional properties should be done.
Line 290 ff: is this classification for studies between brreds rather for substructure of breeds.
Line 295: and0.35, >> and 0.35,
Line 314: and2 (FiguresS3a-b,: please add spaces where appropriate
Conclusions: all concluding remarks are based on visual examination. This is very keen when no statistical evaluation was done. Readers may conclude that your visual approach may have produced this result and a statistical evaluation may have come to other conclusions.
Comments on the Quality of English LanguageMany typos have to be amended.
Author Response
Review 2
1/ Line 28: "and chord diagrams" , please delete.
Answer: Yes, it was deleted.
2/ Line 29: "significant" , please delete, because no statistical tests shown in Results.
Answer: Yes, it was deleted.
3/ Can you please show the average coancestry and inbreeding coefficients for the three Mangalica breeds and their development over birth years. Then you can test whether coancestry and inbreeding were constant or shows a significant trend.
Answer: The evolution of the inbreeding coefficients was shown and discussed in detail in Nguyen et al. 2023. Livest. Sci. 273, 105265 (https://doi.org/10.1016/j.livsci.2023.105265) therefore it is not repeated here.
4/ In addition, you could show confidence intervals for F-ST values
Answer: Confidence interval for pairwise F-ST values between the herds is possible when molecular marker data is available. In our case the calculation was based on pedigree data, so confidence interval could not be determined.
5/ and look at Box-plots and qq-plots if normally distributed. This would also help to evaluate these values.
Answer: Normality of the F-ST values was tested. The distribution was clearly skewed. Density function was placed on every histogram helping to evaluate these values.
6/ Line 35: genealogy, not really done.
Answer: Wright's FST coefficient was calculated based on genealogical data of breeding animals born between 1981 and 2023, with three colour variants (Blonde, Swallow-Belly and Red). These Wright's FST coefficients was analysed using multidimensional scaling to reveal the population substructure.
7/ Line 70-73: rather general; please explain why you conclude that you have a substructure. Farm animals are usually kept in herds. Also give some important results of the demographic analysis which may important for your present study.
Answer: Our intention was to test if there is subdivision in the different Mangalica populations. The related part was reworded.
8/ Line 86: inTable 1. >> in Table 1.
Answer: Thank you! It was corrected.
9/ Line 92: Table1 >> Table 1
Answer: Thank you! It was corrected.
10/ Legend of Table 1 should describe the table contents. please amend.
Answer: Thank you! It was revised.
12/ Line 130 ff: please describe how did you calculate migration rate. Not clear.
Did you calculate number of immigrants minus number of emigrants in the study period divided by the total number of animals being kept in the respective herd with immigration over this period? Please clarify, Did you use weights by herd size?
Checking these rates between breeds and their distributional properties should be done.
Answer: We did not calculate migration rate since the genealogical data lasts for more than 40 years (1981-2023) and one population has 258 herds. Thus, we counted the number of immigrants and calculated the proportion of herd having immigrants and emigrants.
13/ Line 290 ff: is this classification for studies between brreds rather for substructure of breeds.
Answer: “The FST coefficients, which range from 0 to 1, indicate the extent of genetic differentiation….”. As in the reference, this scale can be used for both inter-breed and intra-breed research, please refer to Snegin at al. (2021).
14/ Line 295: and0.35, >> and 0.35,
Answer: Thank you! It was corrected.
15/ Line 314: and2 (FiguresS3a-b,: please add spaces where appropriate
Answer: Thank you! We will read through all the manuscript and fix these typing errors.
16/ Conclusions: all concluding remarks are based on visual examination. This is very keen when no statistical evaluation was done. Readers may conclude that your visual approach may have produced this result and a statistical evaluation may have come to other conclusions.
Answer: According to our knowledge there is no statistical test identifying the number of different groups on the MDS plots. We maintain our opinion that no apparent groups can be seen on these plots. Nevertheless, we provided additional evidence for our conclusion based on the numerical analysis of F-ST values. We checked all F-ST values for every herd from which the largest value for each herd was taken. Then we sorted these (maximum) F-ST values from lowest to highest and calculated the proportion of the total population they represent. Finally, we depicted the results beside the F-ST histograms. It can be seen that the herds having (maximum) F-ST value larger than 0.15 represent negligible part of the population in all breeds.